# Biocontrol Strategies Against Plant-Parasitic Nematodes Using *Trichoderma* spp.: Mechanisms, Applications, and Management Perspectives

**DOI:** 10.3390/jof11070517

**Published:** 2025-07-11

**Authors:** María Belia Contreras-Soto, Juan Manuel Tovar-Pedraza, Alma Rosa Solano-Báez, Heriberto Bayardo-Rosales, Guillermo Márquez-Licona

**Affiliations:** 1Laboratorio de Fitopatología, Subsede Culiacán, Centro de Investigación en Alimentación y Desarrollo, Culiacán 80110, Sinaloa, Mexico; belia.contreras@ciad.mx (M.B.C.-S.); juan.tovar@ciad.mx (J.M.T.-P.); hbayardo123@estudiantes.ciad.mx (H.B.-R.); 2Centro de Desarrollo de Productos Bióticos, Instituto Politécnico Nacional, Yautepec 62731, Morelos, Mexico; asolanob@ipn.mx

**Keywords:** *Trichoderma*, *Meloidogyne*, *Pratylenchus*, *Globodera*, *Heterodera*, plant-parasitic nematodes, biocontrol

## Abstract

Plant-parasitic nematodes represent a significant threat to agriculture, causing substantial economic losses worldwide. Among the biological alternatives for their control, the genus *Trichoderma* has emerged as a promising solution for suppressing various nematode species. This article reviews key studies on the interaction between *Trichoderma* spp. and plant-parasitic nematodes, highlighting the most studied species such as *Trichoderma harzianum*, *Trichoderma longibrachiatum*, *Trichoderma virens*, and *Trichoderma viride*, mainly against the genera *Meloidogyne*, *Pratylenchus*, *Globodera*, and *Heterodera*. *Trichoderma* spp. act through mechanisms such as mycoparasitism, antibiosis, competition for space in the rhizosphere, production of lytic enzymes, and modulation of plant defense responses. They also produce metabolites that affect nematode mobility, reproduction, and survival, such as gliotoxin, viridin and cyclosporine A. In addition, they secrete enzymes such as chitinases, proteases, lipases, and glucanases, which degrade the cuticle of nematodes and their eggs. Furthermore, *Trichoderma* spp. induce systemic resistance in plants through modulation of phytohormones such as jasmonic acid, ethylene, salicylic acid and auxins. The use of *Trichoderma* in integrated nematode management enables its application in combination with crop rotation, organic amendments, plant extracts, and resistant varieties, thereby reducing the reliance on synthetic nematicides and promoting more sustainable and climate-resilient agriculture.

## 1. Introduction

Plant-parasitic nematodes represent a significant threat to global agricultural productivity, as they infect a wide range of economically important crops and cause substantial yield and financial losses [1,2]. Root-knot nematodes (*Meloidogyne* spp.) alone are responsible for yield reductions of up to 60% in tomato, pepper, soybean and other vegetable crops. Cyst nematodes such as *Heterodera glycines* and *Globodera* spp. affect soybean and potato, respectively, leading to regional yield losses of 30–80%. According to global estimates, annual crop losses attributed to plant-parasitic nematodes exceed USD 100 billion. These hidden pests compromise food security and the sustainability of agricultural systems, particularly in tropical and subtropical regions where environmental conditions favor their proliferation [1,3,4,5]. In this context, biocontrol strategies have gained prominence as low-environmental-impact solutions, particularly in response to climate challenges and the growing demands for sustainable agricultural systems [6,7].

Among the available biological alternatives, the use of species from the *Trichoderma* genus has gained relevance due to its effectiveness in the biocontrol of soil-borne pathogens, including plant-parasitic nematodes [7,8]. Various *Trichoderma* species, with their multifunctional capacity, act as natural antagonists that inhibit nematode development through mechanisms such as the production of secondary metabolites, nematotoxic proteins, and competition in the rhizosphere [9].

In addition, *Trichoderma* induces systemic resistance in plants by activating defensive responses and promoting the production of phytoalexins and the expression of defense-related genes [10,11]. This activation increases tolerance to attacks from nematodes and other pathogens, while enhancing the overall health of the plant. The capacity of *Trichoderma* to improve biodiversity and soil structure makes it a valuable tool in the transition toward sustainable agricultural systems that are resilient to climate change.

Climate change causes the gradual increase in soil temperature due to more frequent and prolonged heat waves, exacerbating crop losses [12]. These effects become more severe when nematodes attack plants weakened by thermal stress, altering their physiology and causing greater damage. In this context, tools such as *Trichoderma* spp. gain strategic relevance by contributing not only to nematode biocontrol, but also to mitigating the impact of climate change on agricultural systems.

Several studies have documented the ability of *Trichoderma* spp. to suppress plant-parasitic nematodes through strategies such as direct predation and the induction of resistance in hosts [9,13]. Yan et al. [14] reported that secondary metabolites from *T*. *harzianum* caused high mortality in juveniles of *Meloidogyne incognita* and significantly reduced the hatching of their eggs. These interactions, along with the stimulation of compounds such as jasmonic acid and chitinases in plants, demonstrate the impact of *Trichoderma* on enhancing resistance to nematodes and protecting crops.

This review aims to deepen the understanding of the effects of *Trichoderma* spp. on plant-parasitic nematodes by exploring the nematicidal mechanisms of action and discussing the advantages and limitations of using *Trichoderma*, either alone or in combination with other microorganisms.

## 2. Taxonomy, Morphological Characteristics, and Environmental Adaptation of *Trichoderma* spp.

The genus *Trichoderma* (family Hypocreaceae) has been widely studied, not only for its biotechnological and agricultural importance, but also for its taxonomic complexity; its classification has traditionally been challenging due to the high morphological variability among species. Initially, identifications were based on phenotypic characteristics such as the color and shape of conidia and other reproductive structures. However, classification has advanced considerably with the development of phylogenetic analyses using ITS region sequences, as well as fragments of the *tef-1α* and *rpb2* genes [15]. These molecular analyses have allowed for better differentiation between closely related species and have revealed the existence of cryptic species that could not be discerned by traditional methods [16].

With over 500 recognized species, *Trichoderma* has undergone considerable expansion in its taxonomic classification. The genomic diversity within the genus has been the subject of research, revealing not only the fungus’s adaptability to different environments, but also its ability to hybridize and evolve rapidly in response to ecological pressures. For example, it has been observed that different species exhibit variations in the genes responsible for producing secondary metabolites and enzymes, which confer specific biocontrol properties and organic-matter degradation capabilities [17].

The genus *Trichoderma* is divided into phylogenetic clades, such as Longibrachiatum, Harzianum, Viride, Virens, and Strictipile, defined based on molecular, morphological, and ecological studies. These divisions group species that share genetic and taxonomic similarities. The Longibrachiatum clade includes species associated with specific habitats and is known for containing organisms with a high capacity for producing lignocellulolytic enzymes, making them useful in industrial applications. The Harzianum clade groups species with biocontrol capabilities, with *T*. *harzianum* standing out as one of the most studied and widely used biocontrol agents in agriculture. In the other hand, the Viride clade represents a group of ecologically diverse species capable of colonizing different soil types and exhibiting antagonistic activity against a variety of plant pathogens. The Strictipile clade comprises species that are less commonly studied but have been isolated from soil and decaying wood, and some members have shown potential in biocontrol and plant-growth promotion, although their roles and mechanisms remain less characterized compared to other clades [18].

Species of the genus *Trichoderma* have excellent ecological adaptation capabilities, being able to colonize a wide range of habitats, from forest environments to agricultural soils. This adaptability is due to the fact that these fungi are capable of producing antibiotics, excreting extracellular enzymes, and directly competing for resources, making them excellent competitors for space and nutrients in the rhizosphere. In addition, *Trichoderma* induces the natural systemic resistance in plants and improves nutrient uptake, which also contributes to its success as a rhizosphere colonizer [19]. A central element of its effectiveness as a biological agent is the production of secondary metabolites, such as antibiotics and hydrolytic enzymes (chitinases and glucanases), which inhibit the growth of plant pathogens and decompose lignocellulosic material [20]. The saprophytic capacity of *Trichoderma* allows its establishment in soils rich in organic matter or even in those that have been disturbed, where it can efficiently decompose plant residues and other organic materials [21]. The rapid growth rate of *Trichoderma* spp. is also an important advantage when competing with other microorganisms for space and resources, since the colonization of ecological niches in the rhizosphere is quicker. The ability to compete is enhanced by the production of metabolites that disturb the integrity of other fungi’s membranes and secrete enzymes that degrade the cell walls of pathogens. Likewise, the production of volatile compounds and pigments also contributes to its capacity to compete and survive in diverse environments [22].

The morphology of *Trichoderma* species is not reliable for distinguishing between them. Therefore, phylogenetic identification of species through multilocus sequence typing must be used. Individuals of the genus are characterized by fast-growing and easily sporulating green colonies. They often produce thin mycelium with broad hyphae, branched, with conidiophores irregularly verticillate, which end in clusters of divergent, flask-shaped phialides bearing hyaline conidia in chains. The chlamydospores are formed after seven days or more, are hyaline, and can be intercalary or terminal. From a biological perspective, *Trichoderma* exhibits a versatile lifestyle. It can act as a saprophytic fungus, decomposing organic matter in the soil and actively participating in the carbon cycle [23]. At the same time, other species may display endophytic abilities by colonizing plant tissues without causing apparent damage to the plants [24,25,26]. This endophytic capability is often associated with benefits for the host plants, such as increased resistance to plant pathogens [27].

## 3. Interactions of *Trichoderma* with Plants and Other Microorganisms

*Trichoderma* plays a key role in plant nutrition, due to its ability to interact with the root system and enhance nutrient availability in the soil [28,29,30]. This fungus stimulates root development by producing phytohormones, such as auxins, which increase the absorption surface and optimize the acquisition of essential nutrients [31,32]. Additionally, it facilitates the release of insoluble phosphorus through the production of organic acids (such as citric, oxalic, malic, and lactic acids), which lower the pH of the environment, solubilize calcium, iron, and aluminum phosphates, and release phosphorus in forms available to the plant [33]. The production of phosphatase enzymes and the formation of chelates with blocking cations (Fe^3+^ and Al^3+^) contribute to releasing this key nutrient, enhancing its utilization efficiency. By boosting the absorption of phosphorus, nitrogen, and micronutrients, *Trichoderma* strengthens root growth, resulting in higher photosynthetic efficiency, more vigorous plant development, and increased resistance to adverse conditions [34]. It also promotes a healthy soil microbiome, competing against plant pathogens and fostering synergistic interactions with beneficial bacteria [35].

Due to their complementary action, the interaction between *Trichoderma* and mycorrhizae is crucial for promoting plant growth and improving soil health. Mycorrhizae, particularly arbuscular mycorrhizae, establish a symbiotic relationship with roots that expands the absorption area through a network of extraradical hyphae, enhancing the uptake of nutrients such as phosphorus, nitrogen, and micronutrients [36].

The inoculation of plants with *Trichoderma* and mycorrhizae produces synergistic effects, such as increased root colonization, improved nutrient absorption, and enhanced resistance to abiotic (drought, salinity) and biotic stress (soil-borne pathogens) [37]. The interaction of *Trichoderma* and mycorrhizae strengthens the plants, optimizes nutrient use, and reduces the dependence on chemical fertilizers and pesticides, thereby contributing to more sustainable agriculture. In tomato (*Solanum lycopersicum*), pepper (*Capsicum annuum*), corn (*Zea mays*), and cereals, this strategy has been demonstrated to increase productivity, improve fruit quality, and enhance water and nutrient use efficiency [38].

*Trichoderma* is known for its ability to interact with a wide variety of microorganisms in the soil. These interactions can be both antagonistic and beneficial; they can also promote plant growth by establishing mutualistic relationships or inducing systemic resistance. These complex interactions contribute to *Trichoderma*’s ability to influence the structure and function of soil ecosystems [39]. Several studies have demonstrated the synergism between *Trichoderma* and certain bacteria, among which the application of *T*. *harzianum* in combination with *Bacillus subtilis* promotes growth, yield, and quality in various crops [40]. In peppers, an increase in yield has been reported, associated with the interaction between *Trichoderma* sp., *Pseudomonas fluorescens*, and arbuscular mycorrhizal fungi [41]. In addition, the co-inoculation of *Trichoderma* with bacteria has shown significant benefits for plants by enhancing nutrient supply through atmospheric nitrogen fixation, phosphorus and potassium solubilization, and siderophore production. This synergistic effect can manifest in one or both microorganisms involved [42].

## 4. Application of *Trichoderma* in Nematode Control

The interest in using *Trichoderma* fungi for nematode control began to emerge in the late 1980s, with some of the earliest studies indicating promising nematicidal effects. Windham et al. [43] reported a reduction in the reproduction of *M*. *arenaria* on maize when treated with *T*. *harzianum* and *T*. *koningii*. In 1998, Spiegel and Chet [44] evaluated various isolates of *T*. *harzianum* and *T*. *lignorum* as biocontrol agents against the root-knot nematode *M*. *javanica*, showcasing their nematicidal potential. These foundational studies predate the work of Sharon et al. [45], who demonstrated the effectiveness of *T*. *harzianum* against root-knot nematodes (*Meloidogyne* spp.), and they highlight the early scientific interest in this biocontrol approach. Since then, several studies have confirmed the efficacy of *Trichoderma* spp. in inhibiting plant-parasitic nematodes, with notable effects on the genera *Meloidogyne*, *Pratylenchus*, *Globodera*, and *Heterodera*. More than 60 investigations have reported promising results for the control of these plant-parasitic nematodes, with *M*. *incognita* and *M*. *javanica* being the most extensively studied species (Figure 1).

Figure 1, Figure 2 and Figure 3 were developed based on the information presented in Table 1 and Table 2, respectively, with the aim of graphically representing the collected data.

Nematodes are key soil-borne pests in agriculture, due to their capacity to parasitize a wide range of crops. Root-knot nematodes, especially *Meloidogyne incognita* and *M*. *javanica*, are among the most damaging species globally. These endoparasites induce gall formation in roots, impairing water and nutrient uptake, which leads to stunted growth and significant yield losses [3]. Their high reproductive rate, wide host range, and ability to persist in soil make them difficult to manage through conventional practices. Moreover, their interactions with other pathogens can increase disease severity [46].

The *Trichoderma* clades Harzianum, Viride, and Longibrachiatum have been the focus of studies addressing nematode control (Figure 2), with China, India, and Brazil standing out as leading countries in conducting these studies (Figure 3). The Harzianum and Viride clades are particularly notable for their efficiency in nematode control, owing to their broad genetic diversity and evolutionary adaptability, which allow them to thrive in various soil conditions and target a wide range of nematodes. Those species produce antinematode metabolites (gliotoxin, harzianolide, and viridin) that affect nematode motility, fecundity, and viability. Additionally, they excrete enzymes such as proteases and chitinases to degrade the cuticle and eggs of the nematodes. They are also able to directly parasitize nematode eggs and juveniles via the formation of hyphal networks that immobilize and hydrolyze the nematodes. The effectiveness of root colonization allows them to outcompete nematodes while promoting the presence of favorable microorganisms that assist in their control. In addition, they elicit a defense in plants that deters nematode penetration and feeding, leading to healthy root development. Due to these characteristics, species within these clades have been comprehensively studied and commercialized as biocontrol agents, making them effective for nematode management in diverse agricultural systems.

Among these are the studies conducted by Moo-Koh et al. [47], who reported that *T*. *harzianum*, *T*. *koningiopsis*, *T*. *ghanense*, and *T*. *virens* caused 51 to 100% mortality of J2 juveniles of *M*. *incognita* and *M*. *javanica* in tomato plants. Similarly, other researchers have identified *T*. *virens*, *T*. *hamatum*, *T*. *harzianum*, and *T*. *citrinoviride* as effective agents for inhibiting egg hatching, increasing juvenile mortality, and reducing gall formation in various plant species [14,48,49,50].

Table 1 provides details of the research conducted with *Trichoderma* species against economically important plant-parasitic nematodes, showing that parasitism is the primary mechanism of action identified in most studies. In cases where the specific mode of action was not determined, the term “antagonism” is often used broadly, to describe the overall suppressive effect of *Trichoderma* spp. on nematodes. Spore concentration was the most used application method in these studies, achieving promising results, with egg hatching inhibition and juvenile mortality ranging from 40 to 100%.

**Table 1 jof-11-00517-t001:** Application of *Trichoderma* spp. for the control of plant-parasitic nematodes.

*Trichoderma* Species and Application Type	Mechanism of Action and Study Type	Nematode	Country and Crop	Results and References
*T*. *harzianum* and *T*. *koningi*/C.S	Antagonism/In vivo	*M*. *arenaria*	USA/Corn	Reduction in egg production [43]
*T*. *longibrachiatum*/C.F	Antagonism/In vitro	*Meloidogyne* spp., *Heterodera sachari*, *G*. *rostochiensis*	France	Inhibited movement of infective juveniles [51]
*T*. *harzianum* rifai/-	Parasitism/In vitro	*G*. *rostochiensis*	Pakistan/Potato	*Trichoderma* penetrated cysts and eggs, causing larval death [52]
*T*. *harzianum* and *T*. *lignorum*/C.F	Parasitism/In vivo	*M*. *javanica*	Israel/Tomato	Enhanced plant growth and reduced gall formation [44]
*T*. *virens*/C.F	Antagonism/In vitro and in vivo	*M*. *incognita*	USA/Tomato	42% fewer eggs and J2 per gram of roots [53]
*T*. *harzianum*/C.S	Parasitism/In vitro	*M*. *javanica*	Israel/Tomato	Ability to colonize eggs and second-stage juveniles (J2) [45]
*T*. *harzianum*/C.F	Antagonism/In vitro	*M*. *incognita*	Spain	Significant reduction in eggs [54]
*T*. *atrovirens* and *T*. *harzianum*/C.S	Parasitism/In vitro	*M*. *javanica*	Israel/Tomato	Nematode biocontrol activity [55]
*T*. *harzianum*/C.S	Parasitism/In vitro and in vivo	*M*. *javanica*	Iran/Tomato	Reduces egg hatching and activates defense enzymes [56]
*T*. *asperellum*, *T*. *harzianum*, *T*. *brevicompactum*, *T*. *hamatum* and *T*. *erinaceum*/C.S	Antagonism/In vivo	*M*. *incognita*	Benin/Tomato and carrot	Lowers J2 density, cuts egg production by 86%, and increases tomato yield by 30% [57]
*T*. *harzianum*/C.F	Parasitism/In vitro and in vivo	*M*. *javanica*	Iran/Tomato	Inhibited egg hatching, 84% reduction in egg parasitism, and decreased nematode damage [58]
*T*. *harzianum*/C.S	Parasitism/In vitro and in vivo	*M*. *incognita*	Brazil/Cucumber	Inhibited movement of 60% of eggs and J2 [59]
*T*. *harzianum*/C.S	Antibiosis and induction of resistance of the plant/In vivo	*M*. *enterolobii*	Thailand/Guava	Reduced nematode numbers and stopped J2 development [60]
*T*. *harzianum*/C.S	Parasitism/In vivo	*M*. *javanica*	Saudi Arabia/Tomato	89% of eggs infected; reduced egg hatching by 8.8% and caused 64.5% J2 mortality [61]
*T*. *harzianum*/C.S	Parasitism/In vivo	*G*. *rostochiensis*	Pakistan/Tomato	Cyst wall or egg surface penetration was chemical and mechanical [52]
*T*. *longibrachiatum*/C.S	Parasitism/In vitro and in vivo	*H*. *avenae*	China/Wheat	The parasitic effects of *T*. *longibrachiatum* were >91% after 18 days [62]
*T*. *longibrachiatum*/C.S	Parasitism/In vitro	*M*. *incognita*	China/Cucumber	Strong lethal effect (>88%) and improved plant growth [63]
*T*. *harzianum*/C.S	Antagonism/In vitro	*M*. *incognita*	Mexico/Tomato	Reproduction was reduced by 87–90%, nematode damage and gall formation decreased and plant height and dry biomass increased [64]
*T*. *harzianum* and *T*. *viride*/C.S	Antagonism/In vivo	*M*. *javanica*	Saudi Arabia/Tomato	Suppression of nematode reproduction and gall formation, increased tomato plant growth [65]
*T*. *harzianum*/C.S	Antagonism/In vitro and in vivo	*M*. *incognita*	Ethiopia/Tomato	80% of J2 mortality at 72 h [66]
*T*. *asperellum*, *T*. *harzianum*, *T*. *virens*, *T*. *atroviride*, *T*. *lacuwombatense*, *T*. *viride*/C.S	Antagonism/In vivo	*M*. *hapla*	New Zealand/Tomato	*Trichoderma* strains reduced 1.1 eggs mL soil^−1^ and suppressed galling by 42–88% [67]
*T. harzianum*/C.S	Antagonism/In vivo	*G*. *pallida*	USA/Potato	60% reduction in nematode reproduction [68]
*T. atroviride*/C.S	Induce resistance/In vivo	*M*. *javanica*	Spain/Tomato	Reductions of 42% in galls, 60% in egg masses, and 90% in adult nematodes [69]
*T*. *harzianum*, *T*. *atroviride*, *T*. *virens*/C.S	Antagonism/In vivo	*M*. *incognita*	Mexico/Bell pepper	Egg production reduced by 63% and female production by 14.3%; plant growth enhanced [70]
*T*. *harzianum*/C.S	Induced resistance/In vivo	*M*. *incognita*	Spain/Tomato	Host defenses enhanced during infection, varying by parasitism stage [71]
*T*. *longibrachiatum*/C.S	Parasitism and induced resistance/In vivo	*P*. *brachyurus* and *M. javanica*	Brazil/Soybean	All treatments effectively controlled *P*. *brachyurus* and *M*. *javanica* [72]
*T*. *harzianum*/C.S	Parasitism/In vivo	*M*. *incognita*	Italy/Tomato	Root colonization primed Systemic Acquired Resistance against root-knot nematodes [73]
*T*. *longibrachiatum*/C.S	Parasitism/In vivo	*H*. *avenae*	China/Wheat	89.8% reduction in cysts and juveniles in soil, and 88.3% reduction in J2 and females in roots [74]
*T*. *harzianum*, *T*. *asperellum* and *T*. *longibrachiatum*/-	In vitro	*M*. *javanica*	Morocco/Olive	*Trichoderma* strains killed 50% of the J2s [75]
*T*. *harzianum*, *T*. *hamatum*, *T*. *viride*, *T*. *virens* and *T*. *koningii*/C.S and C.F	In vivo	*M*. *incognita*	India/Tomato	Culture suspensions caused the greatest reduction in hatching and juvenile mortality [76]
*T*. *viride*/C.S	Antagonism/In vivo	*M*. *incognita*	India/Tomato	Increased shoot weight and decreased root weight of tomato, with dose-dependent reductions in galls, egg masses and eggs per egg mass [77]
*T*. *harzianum* and *T*. *viride*/C.S	Antagonism/In vivo	*M*. *incognita*	Pakistan/Tomato	Significant reductions in number of galls, egg masses, eggs per egg mass and reproductive factors of *M*. *incognita* in a dose-dependent manner [78]
*T*. *harzianum*, *T*. *viride*, and *T*. *virens*/C.S	Parasitism/In vivo	*M*. *incognita*	Egypt/Pea	78 to 89% reduction in nematode numbers and gall numbers [79]
*T*. *koningiopsis*/C.S and C.F	Enzymatic hydrolysis/In vitro	*M*. *javanica* and *M*. *incognita*	Brazil	High nematode mortality when applied as an enzymatic filtrate or conidial suspension [80]
*T*. *citrinoviride*/C.S	Antagonism/In vivo	*M*. *incognita*	China/Tomato	Egg hatching inhibition 90% and promoted the growth of tomato plants [48]
*T*. *harzianum*, *T*. *afroharzianum*, *T*. *hirsutum*/C.S	Parasitism/In vitro	*G*. *rostochiensis* and *Meloidogyne* spp.	Algeria/Tomato	Mortality above 70% [81]
*T*. *pseudoharzianum*, *T*. *koingiopsis*, *T*. *asperelloides*, *T*. *afroharzianum*, *T*. *acitrinoviride*, *T*. *hamatum*, *T*. *viride*/C.F	Antagonism/In vitro	*M*. *incognita*	China/Chili	Only the secondary metabolites of *T*. *virens* showed strong nematicidal activity, causing the highest egg hatch inhibition and J2 mortality [82]
*T*. *longibrachiatum*/C.F	Induce resistance/In vitro	*M*. *incognita*	China/Marine algae	The metabolite cyclodepsipeptides 7–9 showed moderate nematicidal activities [83]
*T*. *asperellum* and *T*. *harzianum* (commercial formulates)/C.S	Antagonism/In vitro	*M*. *incognita*	Spain/Tomato and cucumber	The number of egg masses and eggs per plant were reduced. Induced resistance to *M*. *incognita* in tomato but not in cucumber [10]
*T*. *harzianum*/C.S	Parasitism/In vivo	*M*. *incognita*	China/Tomato	Nematode reduction percentage of 62%. The gall number per plant decreased by 75% [14]
*T*. *hamatum*/C.F	Induce resistance/In vitro	*M*. *incognita*	Saudi Arabia/Tomato	Egg hatch inhibition was 78% and juvenile stage mortality rate was 89% [49]
T. *harzianum*, T. *viride* and T. *virens*/C.S	Induce resistance/In vivo	*M*. *javanica*	Egypt/Peanut	The highest percentages reduction in J2 in soil (being 81%) was recorded with *T*. *viride*, followed by *T*. *harzianum* (77%) and *T*. *virens* (73%) [84]
*T*. *asperellum* and *T*. *harzianum*/C.F	Antibiosis/In vivo	*P*. *brachyurus*	Brazil/Soybean	Both isolates have nematicide effects that improve J2 mortality by 41–65% [85]
*T*. *citrinoviride*, *T*. *ghanense*, *T*. *harzianum*, *T*. *koningiopsis*, *T*. *simmonsii*, and *T*. *virens*/C.F	Antibiosis/In vitro	*M*. *javanica* and *M*. *incognita*	Mexico/Tomato	The most lethal strains were *T*. *harzianum*, *T*. koningiopsis, *T*. *ghanense* and *T*. *virens*, which caused 51–100% mortality of J2 of both nematodes [47]
*T*. *virens*/C.S	Antagonism/In vitro and in vivo	*M*. *incognita*	India/Chickpea	Reduction in J2 hatching [50]
*T*. *harzianum*/C.S	Antibiosis/In vivo	*M*. *javanica*	Egypt/Tomato	The penetration rates of nematodes, as well as the number of J2, females, egg mass, and galls were significantly reduced [86]
*T*. *asperellum*/C.S	In vivo	*M*. *incognita*	India/Okra	Hatching suppression 96% and J2 mortality 90% [9]

Note: Abbreviations used in this table: C.S = concentrate of spores; C.F. = culture filtrate.

In addition to these studies, there are effective results from the application of *Trichoderma* spp. in combination with other microorganisms and bioactive substances, as described in Table 2. For example, the control of *Tylenchulus semipenetrans* has been evaluated through the combination of *Trichoderma* spp. isolates and plant extracts, such as neem, karanj, castor oil, and carob galactomannan [87,88]. Also, the effectiveness of *Trichoderma harzianum* combined with 1,3-dichloropropene and organic fertilizers for the control of *M*. *incognita* has been demonstrated [89]. Another strategy to enhance the efficacy of *Trichoderma* spp. in controlling nematodes such as *Pratylenchus brachyurus* and *M*. *javanica* is its combination with bacteria such as *Pseudomonas fluorescens* and *Bacillus* spp. [72,90]. On the other hand, the use of *Trichoderma* in combination with nematophagous bacteria and fungi, such as *T*. *asperellum*, *B*. *subtilis*, and *Purpureocillium lilacinum*, has been proposed, achieving up to an 85% reduction in the nematode reproduction factor [91]. It has also been reported that a mixture of antagonistic fungi, composed of *T*. *harzianum* and *Pochonia chlamydosporia*, applied as a drench in tomato crops, reduced the number of females and egg masses by up to 46% in plants infected by the root-knot nematode *M*. *incognita* [92].

**Table 2 jof-11-00517-t002:** Application of *Trichoderma* spp. consortia with other microorganisms and bioactive substances for the control of plant-parasitic nematodes.

*Trichoderma* Species with Different Products	Mechanism of Action and Study Type	Application Type	Nematode	Country and Crop	Results and References
*T*. *harzianum* + neem, karanj, and castor oil cakes	Parasitism/In vivo	C.S	*Tylenchulus semipenetrans*	India/lime	*Trichoderma* in combination with vegetable oils showed good control of the nematode [87]
*T*. *virens* + *Burkholderia cepacia*	Antagonism/In vivo	C.F.	*M*. *incognita*	USA/Bell pepper	*T*. *virens* suppresses *M*. *incognita*; when combined, it decreases effectiveness [53]
*T*. *harzianum* + *Pseudomonas fluorescens*	Antagonism/In vitro and in vivo	C.F	*M*. *javanica*	Pakistan/Tomato	Mixtures of *P*. *fluorescens* and *T*. *harzianum* improve nematode biocontrol [90]
*T*. *asperellum* and *T*. *atroviride* with Monoclonal and polyclonal antibodies	Parasitism/In vitro	C.S	*M*. *javanica*	Israel/Tomato	*Trichoderma* parasitism increased with antibodies in bioassays [93]
*T*. *longibrachiatum* and cadusafos	Parasitism/In vivo	C.S	*M*. *javanica*	Iran/Zuchini	The optimal concentrations for best plant growth and lowest nematode reproduction were 1.7 mg a.i. kg^−1^ soil and 10^8^ conidia mL^−1^ [94]
*Bacillus licheniformis*, *B*. *subtilis*, *T*. *longibrachiatum*	Parasitism and induced resistance/In vivo	C.S	*P*. *brachyurus* and *M*. *javanica*	Brazil/Soybean	Nematode reduction percentage of 34–40% for *P*. *brachyurus* and 88–92% for *M*. *javanica* [72]
*T*. *asperellum*, *T*. *atroviride*, *Trichoderma* sp. and *Purpureocillium lilacinum*	Antagonism/In vivo	C.S	*M*. *javanica*	Kenya/Pineapple	Reduced nematode egg and egg mass production, lowering root galling damage by 60.8–81.8% and increasing root mass growth [95]
*T*. *viride*, *T*. *harzianum*, *Trichoderma* sp.	Antibiosis/In vitro	C.F	*M*. *incognita* race 2	India/Tomato	Culture filtrates of *Trichoderma* significantly induced inhibition of egg hatching and mortality of *M*. *incognita* race 2 [96]
*T*. *asperellum*, *B*. *subtilis*, *Purpureocillium lilacinum*, and abamectin	Antagonism/In vivo	C.S	*Pratylenchus brachyurus*	Brazil/Soybean	Reduction in the reproduction factor: *T*. *asperellum* 56%, *B*. *subtilis* 78%, and the combination of *T*. *asperellum* with *B*. *subtilis* and/or *P*. *lilacinum* 72.2% [91]
*T*. *harzianum* and *Pochonia chlamydosporia*	Antagonism/In vivo	C.S	*M*. *incognita*	Italy/Tomato	Tomato plants pre-treated with a mixture of beneficial bio-control agents (BCAs), as soil-drenches, were less sensitive to infection of the root-knot nematode [92]
*T*. *harzianum*, *T*. *atroviride*, *T*. *longibrachiatum* and carob galactomannan biopolymer	Antagonism/In vivo	C.S	*M*. *incognita*	Italy/Tomato	Coating tomato roots with the carob galactomannan biopolymer followed by soil application of selected *Trichoderma* strains reduced the root galling index [88]
*Bacillus megatarium*, *B*. *subtilis*, *T*. *harzianum*	Antibiosis/In vivo	C.S and C.F	*M*. *incognita*	India/Sweet basil	Reducing *M*. *incognita* infestation by 46 to 72%. A consortium of BM and TH was the most potent treatment [97]
*T*. *harzianum* and arbuscular mycorrhizae	Antibiosis/In vivo	C.S	*M*. *javanica*	Egypt/Tomato	The lowest number of juveniles was observed in the case of either single mycorrhizal inoculation (45%) or in combination with *T*. *harzianum* (55%) [86]
1,3- dichloropropene with *T*. *harzianum* and an organic fertilizer	Antagonism/In vivo	C.S	*M*. *incognita*	Italy/Tomato	The greatest nematicidal effect was caused by a combination of the three products [89]
*T*. *asperellum*, *T*. *hamatum*, *T*. *atrobruneum*, and *Clonostachys rosea*	In vitro	C.S	*Globodera* spp.	Kenya/Potato	*T*. *asperellum* and *T*. *breve* suppressed nematode egg hatching by 50%, while *T*. *breve* specifically reduced egg viability by 41%
*T*. *harzianum*. and *Bacillus velezensis*	Antibiosis/In vitro	C.S	*M*. *javanica*	Iran/Tomato	Significant nematicidal activity, inhibiting egg hatching (16–45%) and inducing J2 mortality (30–46%) [98]

Note: Abbreviations used in this table: C.S = concentrate of spores; C.F. = culture filtrate.

## 5. Mechanisms of Action of *Trichoderma* Against Nematodes

Several studies detail how *Trichoderma* spp. interact with nematodes, including mechanisms of mycoparasitism, antibiosis, competition for space in the rhizosphere and roots, production of lytic enzymes, and induction of plant resistance [99] (Figure 4).

### 5.1. Parasitism

Parasitism involves the direct interaction between the fungus and the eggs or juveniles of the nematode, inhibiting their development and significantly reducing populations in the soil. One of the key aspects of parasitism is the ability of *Trichoderma* hyphae to firmly adhere to the surface of nematode eggs and juveniles, a process mediated by the recognition of specific molecules present in the cuticle or the external structure of the nematode. This initial adhesion is essential for the establishment of the parasitic attack. Recent studies have shown that this hyphal adhesion is mediated by specific carbohydrate–lectin interactions. For instance, *Trichoderma asperellum* conidia can bind to the surface of *Meloidogyne javanica* eggs and juveniles through Ca^2+^-dependent lectin–carbohydrate recognition. This process may be further stabilized by antibody-like binding to both conidia and nematodes, facilitating firm attachment [55,93]. Additionally, Inbar and Chet [100] reported the presence of lectins in culture filtrates of *T*. *viride* and *T*. *harzianum*, suggesting that these proteins play a significant role in the recognition of, and attachment to, host surfaces. These molecular interactions are considered a crucial step in the parasitic process, enabling the fungus to remain anchored and initiate penetration. From there, the fungus envelops the eggs through the growth of its hyphae and generates an extracellular matrix that facilitates colonization and the subsequent development of specialized structures, such as appressoria, which exert mechanical pressure on the outer wall of the egg, promoting its penetration and the subsequent invasion of its internal contents [101].

In parallel, *Trichoderma* spp. complement the mechanical attack with a potent biochemical mechanism that enhances the infection process’s effectiveness. This biochemical arsenal is mainly regulated by heterotrimeric G protein, cAMP, and MAPK motif signals, based on the production of hydrolytic enzymes with nematicidal properties, such as chitinases, proteases, lipases, and glucanases. These enzymes break down vital components of the cuticle and cell wall of nematode eggs and juveniles, compromising their structural integrity and allowing fungal hyphae to penetrate and, ultimately, kill them. Furthermore, these enzymes disrupt the nematodes’ capacity to infect plant roots [14,45,62].

The combined effects of mechanical disruption and enzymatic breakdown significantly diminish the survival rate of eggs and limit juvenile hatching, thus disrupting the nematode life cycle. Recent research indicates that this combination of physical and biochemical mechanisms can greatly reduce the population density of plant-parasitic nematodes, positioning *Trichoderma* spp. as a promising element in integrated nematode management strategies within agricultural systems [49].

### 5.2. Secondary Metabolite Production (Antibiosis)

*Trichoderma* spp. produce secondary metabolites with nematicidal activity that act through various mechanisms. The production of these metabolites is influenced by environmental and physiological factors, including cultivation conditions such as medium composition, pH, temperature, aeration, and nutrient availability [82,102]. Among these metabolites, one of the most relevant groups is the volatile organic compounds (VOCs), including alcohols, ketones, terpenes, and other organic molecules, which can affect the mobility, reproduction, and survival of nematodes, thereby contributing to their control [103].

In addition to VOCs, *Trichoderma* also synthesizes non-volatile metabolites, such as peptaibols, alkaloids, and polyketides, which impact the nervous system, reproduction, and mobility of nematodes [104,105,106].

Moreover, the interaction of *Trichoderma* spp. with nematodes or whit plant root exudates released in response to nematode infection can trigger the production of nematicidal secondary metabolites [8]. Several studies have reported that the secondary metabolites produced by certain *Trichoderma* species directly affect egg hatching and juvenile survival of *M*. *incognita*. These compounds include toxic molecules such as gliotoxin, immunosuppressive peptides like cyclosporin A, and other bioactive secondary metabolites, which exert nematicidal or nematostatic effects by interfering with the development and viability of nematode populations [51,82,107,108].

### 5.3. Competition for Resources and Rhizosphere Colonization

*Trichoderma* controls plant-parasitic nematodes through competition for resources and rhizosphere colonization, displacing them by occupying ecological niches in the roots. This fungus adheres to and colonizes the roots using specialized structures such as hyphae and appressoria, preventing nematodes from finding feeding or reproduction sites [28]. In addition, it competes for root exudates, a key nutrient source for nematodes, reducing their availability and limiting their development [109]. By occupying the same sites that nematodes would use to parasitize the roots, *Trichoderma* acts as a physical barrier, blocking their access [110]. At the same time, it induces changes in the soil microbiome, stimulating antagonistic microorganisms that produce nematicidal compounds or enzymes that degrade nematode eggs and juveniles, such as *Pseudomonas fluorescens*, *Bacillus subtilis*, *Purpureocillium lilacinum*, and *Pasteuria penetrans* [82,90,111,112]. It also secretes secondary metabolites, such as organic acids and volatile compounds, which alter the composition of the microbiome, creating an unfavorable environment for nematodes [108,113].

### 5.4. Induction of Systemic Resistance in Plants

The induction of systemic resistance is one of the key mechanisms by which *Trichoderma* protects plants against plant-parasitic nematodes, activating their natural defenses through the production and regulation of hormones such as jasmonic acid, ethylene, and salicylic acid, which strengthen physical and chemical barriers against infection [28,109].

This defense response may occur through two main pathways: Induced Systemic Resistance (ISR), typically triggered by beneficial microbes like *Trichoderma* via jasmonic acid and ethylene signaling, and Systemic Acquired Resistance (SAR), usually activated by pathogen infection and mediated by salicylic acid [28]. In the context of ISR, *T*. *harzianum* colonization has been shown to reduce *M*. *incognita* infection in tomato by priming the plant’s defense genes [71]. Although less frequently reported, SAR may also be enhanced indirectly. In addition, plants recognize conserved microbial patterns known as Pathogen-Associated Molecular Patterns (PAMPs) or Microbe-Associated Molecular Patterns (MAMPs), such as chitin fragments, small proteins, or polysaccharides produced by *Trichoderma*. These molecules trigger basal immune responses without causing disease [114]. When nematodes damage root cells, they release Damage-Associated Molecular Patterns (DAMPs), which further amplify plant defenses. These overlapping signals collectively contribute to enhanced nematode resistance [109]. Jasmonic acid promotes the synthesis of defensive compounds, such as protease inhibitors and secondary metabolites toxic to nematodes [28,109]. Ethylen promotes the lignification of cell walls, reinforcing the tissue structure and hindering the penetration and feeding of nematodes [115]. Meanwhile, salicylic acid stimulates the accumulation of phytoalexins and the activation of resistance genes, negatively affecting the nematodes [104]. As a result, plants treated with *Trichoderma* spp. show reduced penetration and reproduction of nematodes, a decrease in cyst and egg formation in the case of *Globodera* spp. and *Heterodera* spp. [62], as well as a significant reduction in the formation of galls in roots infected by *Meloidogyne* spp. [28], and improved growth and vigor, demonstrating its ability to reduce the damage caused by these plant-parasitic nematodes and enhance stress tolerance [109]. Additionally, molecular studies have shown that plants treated with *Trichoderma* spp. exhibit higher expression of defense genes, further strengthening their defense response against nematodes [115].

To provide a clearer overview of the diverse strategies employed by *Trichoderma* spp., Table 3 summarizes the main mechanisms of action involved in the control of plant-parasitic nematodes and plant pathogenic fungi. These mechanisms include the production of secondary metabolites, secretion of lytic enzymes, induction of systemic resistance, physical interactions, and competition for space and nutrients. Although many of these strategies are shared between both target groups, there are relevant distinctions. For instance, direct parasitic interaction, commonly referred to as mycoparasitism in the context of fungal pathogens, has also been reported in nematodes, where *Trichoderma* hyphae adhere to eggs or juveniles, form appressoria, and penetrate the host through mechanical and enzymatic means. This comparative analysis highlights the versatility and adaptability of *Trichoderma* spp. in suppressing a wide range of plant pathogens, reinforcing its potential as a valuable component of integrated pest- and disease-management programs.

## 6. Commercial Applications, Limitations, and Future Perspectives of *Trichoderma* spp. in Nematode Management

Numerous studies have documented the efficacy of commercial formulations of *Trichoderma* spp. in the control of plant-parasitic nematodes. For instance, preparations based on *T*. *harzianum* have been shown, both in greenhouse and field conditions, to significantly reduce galling index and egg production of *M*. *javanica* and *M*. *incognita*, while improving plant vigor and yield in crops such as tomato and soybean [44]. In one specific study, soil treatment with *T*. *harzianum* filtrate reduced the gall index from 2.78 to 0.4 in celosia plant infected with *M*. *incognita*, along with a drastic reduction in nematode population and notable improvements in plant growth [122]. Additionally, commercial formulations containing strains T-22 (*T*. *harzianum*) and T-34 (*T*. *asperellum*) induced systemic resistance in tomato plants against *M*. *incognita*, showing a cumulative effect, with resistance genes such as *Mi-1.2* [10].

The management of plant-parasitic nematodes using *Trichoderma* faces several challenges. Variability in effectiveness is influenced by the specific species or strain used, the target nematode, soil physicochemical properties, and interactions with native-soil microbiota. Factors such as formulation stability, storage, handling, and application methods significantly impact the viability and efficacy of these products [123]. Moreover, environmental conditions (temperature, moisture, pH) and interactions with other agrochemicals can affect fungal survival and biocontrol activity. The lack of standardized protocols regarding inoculation techniques, dosage, and environmental requirements further limits consistent field application, particularly in smallholder contexts. Addressing these issues through targeted research and regulatory support is essential to fully harness *Trichoderma* as a reliable nematode management tool. Figure 5 presents a proposed diagram summarizing the management of *Trichoderma* spp. from isolation to field application.

The potential to combine *Trichoderma*-based biopreparations with chemical nematicides presents a promising strategy within integrated pest management (IPM). Evidence suggests additive or synergistic effects that allow reduced chemical inputs while maintaining efficacy, thus minimizing environmental impact. Some *Trichoderma* strains tolerate low concentrations of pesticides, and co-application may enhance root colonization and biocontrol performance [89]. Advances in genetic engineering offer exciting opportunities to improve *Trichoderma* strains. Current research focuses on overexpressing genes related to secondary metabolite production, plant immune activation, and stress tolerance, as well as applying CRISPR-Cas9 for precise genome editing. These approaches aim to develop more robust strains capable of functioning effectively under diverse and challenging field conditions [124].

It is crucial to consider biotic and abiotic factors that influence the performance of *Trichoderma* formulations. Soil texture, organic matter content, microbial competition, and environmental fluctuations can significantly alter fungal survival and efficacy. Laboratory results often differ from field outcomes, due to the complexity and variability of natural ecosystems [125]. Therefore, extensive field validation and long-term trials are necessary to ensure the practical reliability of these products under real agricultural conditions. Looking forward, *Trichoderma* spp. can play a central role in integrated nematode management strategies, provided current knowledge gaps are addressed. Omics technologies (genomics, transcriptomics, proteomics, and metabolomics) promise to enhance understanding of the complex interactions between *Trichoderma*, nematodes, other soil microorganisms, and host plants [126]. This knowledge will enable the identification of key genes and pathways to select or engineer superior strains and optimize application protocols. Furthermore, microbial consortia combining *Trichoderma* spp. with other beneficial microbes, such as plant growth-promoting rhizobacteria (PGPR), have the potential to improve biocontrol efficacy, product stability, and persistence in the soil [127]. Finally, integrating *Trichoderma* spp. with cultural practices (crop rotation, organic amendments, etc.) and host genetic resistance can reduce reliance on synthetic nematicides, supporting more sustainable and environmentally friendly agricultural-production systems.

## 7. Conclusions

The effectiveness of *Trichoderma* spp. in controlling plant-parasitic nematodes from the genera *Heterodera*, *Globodera*, *Meloidogyne*, and *Pratylenchus* has been widely demonstrated. The biocontrol capacity of *Thichoderma* against plant-parasitic nematodes arises from a combination of diverse mechanisms, including the production of enzymes that facilitate the infectious process, metabolites with antimicrobial activity, its rapid growth, which allows it to compete for space and resources, and the induction of systemic resistance in plants. However, the efficacy of a treatment based on *Trichoderma* can be affected by multiple factors, for example, the strain selected, the nematode, the application technique, and soil conditions. This highlights the need for further studies to optimize the extensive use of *Trichoderma*. The recent development of formulations incorporating *Trichoderma* spp. as a part of microbial consortia is a promising alternative to the conventional use of individual strains, to increase their effectiveness and sustainability. As research expands on the long-term ecological consequences of using *Trichoderma* in agriculture, and as protocols for its use are developed, we will be able to recommend it for extensive and widespread applications in a variety of sustainable agricultural systems. The adaptability and flexibility of *Trichoderma* spp. make them a promising asset for the development of sustainable alternatives to nematode management.

## Figures and Tables

**Figure 1 jof-11-00517-f001:**
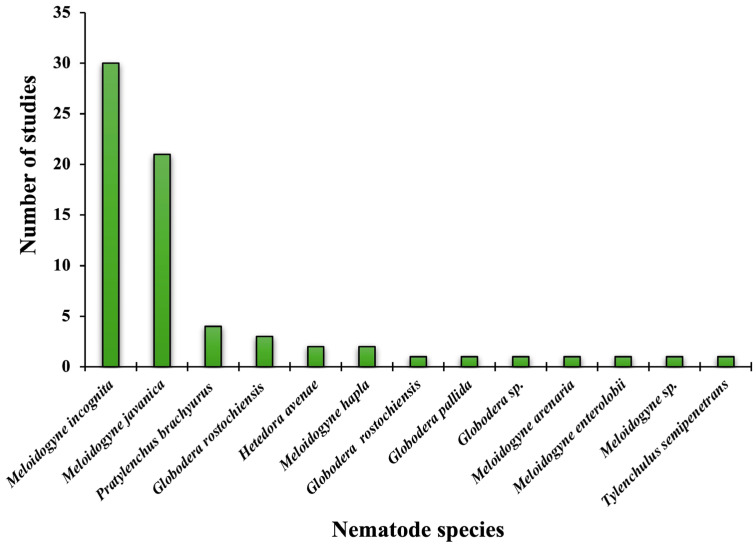
Frequency of studies on the biocontrol of plant-parasitic nematodes with *Trichoderma* spp.

**Figure 2 jof-11-00517-f002:**
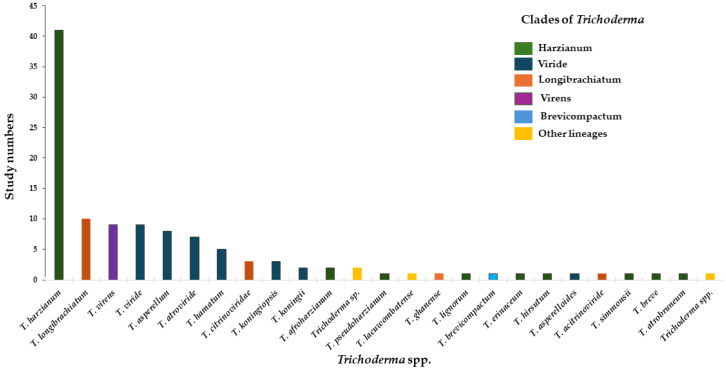
Clades of *Trichoderma* most reported as biocontrol agents against plant-parasitic nematodes.

**Figure 3 jof-11-00517-f003:**
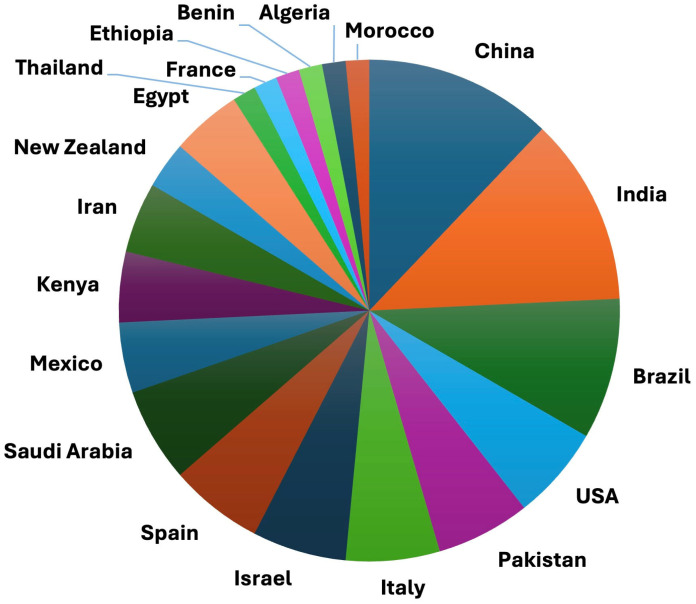
Countries where studies on *Trichoderma* spp. as biocontrol agents for nematodes have been conducted.

**Figure 4 jof-11-00517-f004:**
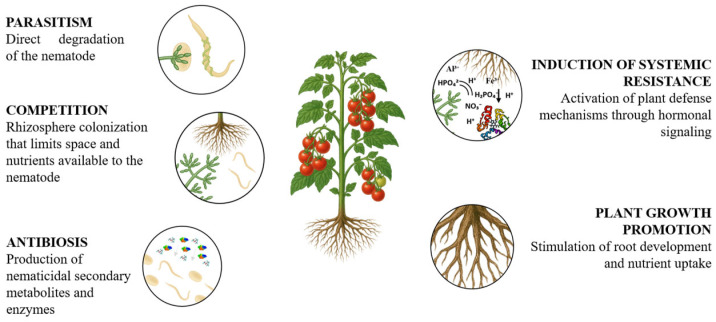
Mechanisms of *Trichoderma* in the control of plant-parasitic nematodes and in the promotion of plant growth and stress tolerance.

**Figure 5 jof-11-00517-f005:**
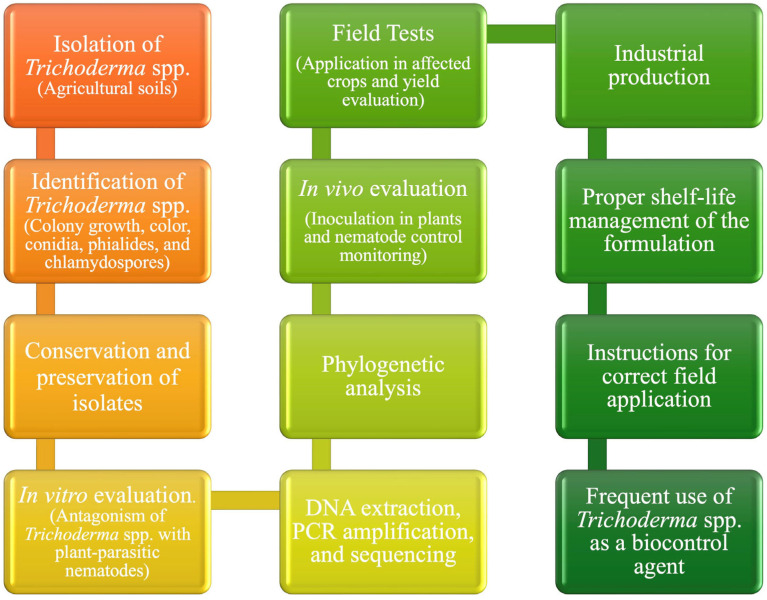
Isolation and application of *Trichoderma* spp. with a farmer-friendly approach.

**Table 3 jof-11-00517-t003:** Comparison of *Trichoderma* spp. mechanisms of action against plant-parasitic nematodes and fungal pathogens.

Mechanism of Action	Against Nematodes	Against Fungal Pathogens
Production of secondary metabolites	Gliotoxin, viridin, cyclosporin A, acetic acid—inhibit egg hatching and juvenile development [48,51,107,116]	Peptaibols, gliotoxin, 6-pentyl-α-pyrone—inhibit fungal growth and spore germination [117]
Production of lytic enzymes	Proteases, chitinases—degrade cuticle or eggshell [49,78,118]	Chitinases, glucanases—degrade fungal cell walls [82]
Induced systemic resistance (ISR)	Activation of jasmonic acid/ethylene pathways—increased plant-defense compounds [10]	Similar activation to ISR—enhanced plant resistance to fungal infection [10,119]
Direct physical interaction	Limited or absent [45]	Mycoparasitism: coiling, penetration, and degradation of fungal hyphae [28]
Competition for space and nutrients	Present in the rhizosphere [19]	Strong competition on root and rhizoplane surfaces [120,121]
Plant growth promotion	Enhances plant tolerance to nematode stress [10,65]	Improves plant vigor, indirectly reducing fungal susceptibility [114]

## Data Availability

No new data were created or analyzed in this study. Data sharing is not applicable to this article.

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
