# Peer review of "Biocontrol Strategies Against Plant-Parasitic Nematodes Using Trichoderma spp.: Mechanisms, Applications, and Management Perspectives"

_jof, 2025, doi:10.3390/jof11070517_

Round 1

Reviewer 1 Report

The article titled: "Biocontrol strategies against plant-parasitic nematodes using Trichoderma spp.: Mechanisms, applications, and management perspectives" is a very good review of this specific topic of biocontrol of nematode by using Trichoderma spp. The article is well written and easy to follow and cover adequately the different topic of this research. Nevertheless, this reviewer will recommend to consider the following comments and suggestions:

Line 98. Authors should describe briefly the clade Strictipile of Trocherma.

Line 191: Authors should incorporate a briefly description of the importance of nematode in agriculture, especially the two species mentioned in line 185.

Line 324: “…produce defensive com-pounds that are toxic to nematodes, such as …” Pleae, mention those compounds. The same in line 330

Lines 462 and 463: “… Omics sciences, for example, can provide us with valuable information about the interactions between Trichoderma, nematodes, other microorganisms, and plants.

Minor details are decribed in the section of major comments. 

Reviewer 2 Report

The information contained in the manuscript is consistent with the current state of knowledge, but is predominantly descriptive. The authors cited over 60 publications on the control of pathogenic nematodes using fungi of the genus Trichoderma and other microorganisms or natural substances in various countries. Based on these sources, they have developed three charts on the most commonly controlled nematode species, the most frequently used Trichoderma species, and the countries where this biocontrol agent has been used. Unfortunately, the description of the key mechanisms of action of Trichoderma spp. against nematodes, as well as their application in agricultural practice, is insufficient. The authors do not critically refer to the cited studies and fail to discuss their findings. There is no comparison between the mechanisms of nematode control and fungal pathogens. Many points are repeated – some could be combined and shortened, while others require more in-depth explanation. One gets the impression that the authors may have relied primarily on abstracts rather than thoroughly reviewing the full content of the cited articles. They focus heavily on describing the taxonomy and morphology of Trichoderma fungi, as well as the general characteristics of biopreparations (e.g., their application methods and benefits), which are already well known and do not contribute new insight, thereby diminishing the scientific value of the article.

The manuscript submitted for review provides a comprehensive review of the literature on the mechanisms of action of Trichoderma fungi and a compilation of studies on their use against plant nematodes. The topic is important and timely. However, the manuscript has several shortcomings that need to be addressed in order to meet the standards of a journal such as the Journal of Fungi.

Title. The title raises no objections.

 Abstract

  1. What do the authors mean when they write that one of the mechanisms of action of Trichoderma fungi, alongside antibiosis, lytic enzyme secretion, and competition, is antagonism?
  2. The authors should organize their knowledge of the mechanisms of action of Trichoderma fungi that are based on antagonism. Antagonistic activity may be directed against the pathogen (e.g., competition, antibiosis, lytic enzymes, parasitism, hyperparasitism) or indirectly benefit the plant (e.g., biofertilization, growth stimulation).

 Please read the sample articles.

Reference 26: Harman, Gary E., et al. "Trichoderma species-opportunistic, avirulent plant symbionts." Nature reviews microbiology 2.1 (2004): 43-56.

Reference 30: Woo, Sheridan L., et al. "Trichoderma-based products and their widespread use in agriculture." Open Mycol. J 8.1 (2014): 71-126.

Sharma, Vivek, Richa Salwan, and P. N. Sharma. "The comparative mechanistic aspects of Trichoderma and probiotics: scope for future research." Physiological and Molecular Plant Pathology 100 (2017): 84-96.

TyÅ›kiewicz, Renata, et al. "Trichoderma: The current status of its application in agriculture for the biocontrol of fungal phytopathogens and stimulation of plant growth." International Journal of Molecular Sciences 23.4 (2022): 2329.

1. Introduction

Line 35. This point should be elaborated. What losses do nematodes cause, what key crops do they affect, and what are the financial losses? See e.g. Reference 1.

2. Taxonomy; 3. Ecological Adaptation and Niches, 4. Morphological and Biological Characteristics

1. Points 2–4 can be consolidated into a single section, e.g.: Taxonomy, morphological characteristics of Trichoderma spp. and environmental adaptation. This is common knowledge and does not add anything new to the article.

2. When describing the morphology of the colony and cell, photos can be included to make the article more attractive.

A) Siddiquee, Shafiquzzaman, and Shafiquzzaman Siddiquee. "Morphology-based characterization of Trichoderma species." Practical Handbook of the Biology and Molecular Diversity of Trichoderma Species from Tropical Regions(2017): 41-73.

B) Cai, Feng, et al. "The current state of Trichoderma taxonomy and species identification." Advances in Trichoderma biology for agricultural applications. Cham: Springer International Publishing, 2022. 3-35.

5. Interactions of Trichoderma with Other Microorganisms

Line 136–149 do not address the interactions between Trichoderma and microorganisms, but rather between Trichoderma and plants. Therefore, it should be relocated to section 3. “Ecological....”  if retained at all. Its removal should be considered.

6. Application of Trichoderma in Nematode Control

1. Antagonistic fungi of the genus Trichoderma have been known in the scientific literature since the 1930s, when the first publications by Weindling (1934, 1936) appeared. The interactions between Trichoderma and fungal pathogens are well documented, and Trichoderma-based products are commercially applied in various crops. However, the manuscript does not address a key question: When did the interest in using Trichoderma fungi for nematode control emerge, and who were the first researchers to explore this application?

Weindling R.: Studies on the lethal principle effective in the parasitic action of Trichoderma lignorum on Rhizoctonia solani and other fungi. Phytopathol. 24, 1153 - 1179 (1934).

Weindling R., Emerson O.H.: The isolation of a toxic substance from the culture of a Trichoderma. Phytopathol. 26, 1068 - 1070 (1936).

2. Line 184. Please indicate that these are publications in Tables 1 and 2.

3. It should be noted that Figs. 1–3 were created based data presented in Tables 1 and 2. This connection must be made explicit to ensure clarity for the average reader. Consider whether publications below 5 are relevant; perhaps the others should be marked as, for example, Fig. 1 for Globodera spp., etc. Similarly, in the case of Fig. 2, it should be rewritten. Set the values in descending order, not in clades.

Fig. 3 contains a duplicate legend.

Tables 1 and 2 are difficult to read in their current form and should be reformatted for clarity, e.g. Trichoderma spp and application type can be combined into one column, similarly Country and crop, results and reference.

Line 245. „of action are parasitism and antagonism”.  See Abstract.

7. Mechanisms of Action of Trichoderma against Nematodes

Lines 272-275. According to the authors, what does antagonism refer to?

7.1 Parasitism

Parasitism is a key mechanism of action of Trichoderma fungi on nematodes. It should be supported by some photos and a detailed description. How does this mechanism differ from the control of pathogenic fungi? Which CWDEs enzymes play a key role here?

Line 292. „of specific molecules present in the cuticle”. This needs to be expanded upon.

7.2 Secondary Metabolite Production (Antibiosis)

In addition to parasitism, antibiosis is a key mechanism of action of Trichoderma fungi in controlling nematodes, based not only on volatile but also on non-volatile metabolites. A more detailed discussion is needed, supported by examples from the literature. Which groups of metabolites exhibit nematicidal activity, and which are more effective against pathogenic fungi?

Line 328-330. Additionally, the interaction with nematodes or their exudates can induce the production of nematicidal secondary metabolites by Trichoderma spp. [104]. Please cite what research tests were used and what metabolites were examined.

In the abstract (line 20), the authors mention “paralysis” as one of the mechanisms of action of Trichoderma fungi. However, this mechanism is not elaborated upon in the main text of the manuscript. The authors should clarify whether paralysis refers to an effect related to antibiosis? This is not the same as parasitism, mycoparasitism, or hyperparasitism.

7.3 Competition for Resources and Rhizosphere Colonization

Line 344 – 345. Please expand this paragraph.

At the same time, it induces changes in the soil microbiome, stimulating antagonistic microorganisms that produce nematicidal compounds or enzymes that degrade nematode eggs and juveniles”. What microorganisms other than Trichoderma fungi?

 Line 350-358 concern the induction of immunity, see point 7.4.

7.4 Induction of Systemic Resistance in Plants

The authors describe the key mechanism succinctly, without distinguishing between ISR, SAR, PAMPs/MAMPs/DAMPs. The relevant articles are cited, but there are no relevant examples. Please see to the yours references.

8. Characteristics of commercial products based on Trichoderma spp.

The authors refer to biopreparates used against fungal pathogens but do not provide any examples of commercial products based on Trichoderma fungi for controlling nematodes. In other words, while research is ongoing, the results are not being translated into commercial applications. This issue should be addressed.

This section does not add significant value, as it presents information that is already well known.

9. Limitations and Challenges in the Use of Trichoderma and 10. Future Perspectives

Points 8, 9 and 10 should be combined. Please describe the potential for using such biopreparations in combination with chemical pesticides, discuss ongoing research in the field of genetic engineering, and elaborate on the influence of biotic and abiotic factors as well as differences between laboratory and field conditions.

Round 2

Reviewer 2 Report

The manuscript describes the interactions between Trichoderma spp. and mainly plant pathogenic fungi as well as nematodes. This is a widely known topic that is often described in the literature on the subject.

The authors responded to the most of my earlier comments. However, some shortcomings are still visible. Unfortunately, I have still some comments for Authors before acceptance for publication.

The authors mistake the mechanism of antibiosis with induction of systemic resistance as well as mycoparasitism. Trichoderma fungi produce secondary metabolites with antibiotic properties. They then act lethal and/or inhibit microorganisms or nematodes. Secondary metabolites also include lytic enzymes (e.g., chitinases, glucanases) that act in the parasitic process. Another group are protein metabolites, that take part in induce systemic resistance. 

This needs to be clarified.

 Line 24-26. Induction of systemic resistance should not be linked to antibiosis, but rather to the production of phytohormones such as auxins eg. IAA, IBA.

Moreover, fungi from genus Trichoderma do not produce acetic acid as a primary metabolite.

Line 304 – 305. “induced resistance, as well as through plant defense induction”. Isn't it the same thing? Please explain.

Line 315-316. Figure 4 refers to the interaction between Trichoderma spp. and fungal pathogens, not between Trichoderma spp. and nematodes. Please modify the diagram to show the interactions between Trichoderma, nematodes, and plants.

What program was used to create figure 4?

Line 355-383. Point 5.2. needs to be reworded.

Line 369-371 should be placed after the first sentence (line 356-357).

Line 363 - 368 refers to the induction of systemic resistance, not antibiosis.

In Tables 1 and 2, the conjunction “e” is used instead of “and” between in vivo and in vitro.

Line 280 and 300C.F. = culture filtration” or filtrate.

„S.C. = spore concentration” or concentrate of spores. Please explain.

In Table 1, you can combine columns 2 and 5. Similarly in Table 2.

Please restore the earlier name of column 2 i.e. “Mechanism of action”.

The numbering of references should be checked throughout the manuscript.

Please remove any spaces between initials.
